# Characterization of Triterpene Saponin Glycyrrhizin Transport by *Glycyrrhiza glabra*

**DOI:** 10.3390/plants11091250

**Published:** 2022-05-05

**Authors:** Kakuki Kato, Asako Horiba, Hiroaki Hayashi, Hajime Mizukami, Kazuyoshi Terasaka

**Affiliations:** 1Graduate School of Pharmaceutical Sciences, Nagoya City University, 3-1 Tanabe-Dori, Mizuho-Ku, Nagoya 467-8603, Japan; tunatuna80doradora@true.ocn.ne.jp (K.K.); a.horiba@nexus115.co.jp (A.H.); hajimem@makino.or.jp (H.M.); 2College of Pharmaceutical Sciences, Ritsumeikan University, 1-1-1 Noji-Higashi, Kusatsu 525-8577, Japan; hhayashi@fc.ritsumei.ac.jp

**Keywords:** glycyrrhizin, triterpene, ABC transporter, H^+^-symporter

## Abstract

Glycyrrhizin (GL), a triterpene compound produced by *Glycyrrhiza* species, is a crucial pharmacologically active component of crude drugs. In contrast to the biosynthesis of GL in plants, little is known about GL transport and accumulation in plants. The transport mechanism of GL was characterized using cultured cells of *Glycyrrhiza glabra*. Cultured cells of *G. glabra* efficiently incorporated exogenously supplied GL. Proton pump inhibitors, such as probenecid and niflumic acid, as well as a protonophore (carbonylcyanide *m*-chlorophenylhydrazone), markedly inhibited GL uptake by cultured cells, whereas vanadate exhibited a moderate inhibition. Furthermore, GL transport by *G. glabra* tonoplast vesicles is dependent not on a H^+^-electrochemical gradient but MgATP and is markedly inhibited by vanadate. These results suggest that GL uptake by cultured cells is mediated by a H^+^-symporter in the plasma membrane and an ATP-binding cassette transporter, which has high specificity for the aglycone structure of GL on the tonoplast.

## 1. Introduction

Triterpene saponins are widely distributed in plants and accumulate in various medicinal plants as bioactive compounds [1]. The licorice, dried roots and stolons of *Glycyrrhiza* plants (*G. uralensis* Fisch. and *G. glabra* L., Fabaceae) are among the most important crude drugs in the world [2] and contains large amounts (2–8% of the dry weight) of glycyrrhizin (GL), an oleanane-type triterpene saponin.

Because of its sweet taste, GL is used worldwide as a natural sweetener and flavoring additive [3]. Furthermore, GL demonstrates various pharmacological activities, such as anti-inflammatory, immunomodulatory, antiulcer, and antiallergy activities [4,5,6,7]. Moreover, GL shows an antiviral activity against various DNA and RNA viruses, including HIV and severe acute respiratory syndrome (SARS)-associated coronavirus [8,9,10]. In Japan, GL has been used for more than 20 years as a hepatoprotective agent for chronic hepatitis [11,12]. Therefore, a large amount of licorice and its extracts are available worldwide as sweetening agents and medicinal materials.

The biosynthesis of GL has been extensively studied, and it involves an initial cyclization of 2,3-oxidosqualene, a common precursor of both triterpenes and phytosterols [13], leading to the formation of a triterpene compound, β-amyrin, followed by a series of oxidative reactions at positions C-11 (two-step oxidation) as well as C-30 (three-step oxidation) and bonding of two molecules of glucuronic acid at the C-3 hydroxyl group (Figure 1). Two genes encoding enzymes, namely squalene synthase and β-amyrin synthase (bAS), involved in the early stages of triterpene skeleton formation were functionally isolated from *G. glabra* [14,15]. Recently, in vitro and in vivo (in bAS-expressing yeast) studies have shown that CYP88D6 and CYP72A154 isolated from *G. uralensis* using an expressed sequence tag library catalyzed two sequential oxidation steps in the GL biosynthetic pathway, i.e., the conversion of β-amyrin to 11-oxo-β-amyrin via 11α-hydroxy-β-amyrin and 11-oxo-β-amyrin to glycyrrhetinic acid, respectively [16,17]. In addition, two glycosyltransferases involved in the glucuronidation of glycyrrhetinic acid were also identified [18,19]. Although GL is economically important, various efforts undertaken for its biotechnological production using plant cell and tissue cultures have been unsuccessful to date. Neither cell suspension cultures nor hairy root cultures of *G. glabra* produced detectable amounts of GL [20,21]. In contrast to the extensive investigation on the biosynthesis and production of GL, intercellular and intracellular transport of GL has yielded less research findings, except that GL is considered to be accumulated in vacuoles.

Recently the accumulation and membrane transport of plant secondary metabolites were examined [22,23]. Understanding the intercellular and intracellular transport of secondary products is essential for elucidating the physiological functions of small-molecule organic compounds and for metabolic engineering to increase the productivity of valuable secondary metabolites. Triterpenoids comprise many pharmacologically important compounds and are targets for metabolic engineering. However, no study has described the transport mechanism of triterpenoid compounds.

In this study, we characterized for the first time the transport mechanism of GL by using cultured cells and tonoplast vesicles of *G. glabra*. In cultured cells, GL uptake in the plasma membranes was coupled with a H^+^-electrochemical gradient; however, in tonoplast vesicles, GL transport was stimulated by MgATP and inhibited by vanadate, but not by NH_4_^+^ and bafilomycin A_1_, which suggests that uptake involved an ABCC-type ATP-binding cassette (ABC) transporter.

## 2. Results

### 2.1. GL Uptake by Cultured Cells of G. glabra

GL was added to the cell suspension cultures of *G. glabra* at the early growth phase (a week after subculture) at a final concentration of 100 µM. The GL concentration in the cell and medium was periodically determined for 6 h after GL addition. The loss of GL from the medium during incubation was considered the amount taken up by the cells (Figure 2 and Appendix A). GL was found to be stable in the medium without cells over a 6 h incubation period. GL uptake did not affect cell viability throughout the 24 h incubation period.

Furthermore, we examined GL uptake by cultured cells of *Lithospermum erythrorhizon* and *Catharanthus roseus*, which are GL-nonproducing species. Neither of them exhibited a significant GL uptake (data not shown).

### 2.2. Characterization of GL Uptake by Cultured Cells of G. glabra

The effects of various transport inhibitors on GL uptake by *G. glabra* cells were examined (Figure 3). These compounds did not affect cell viability at the concentrations used in the present investigation. Probenecid and niflumic acid, major anion blockers, strongly inhibited GL uptake. In addition, carbonylcyanide *m*-chlorophenylhydrazone (CCCP), a protonophore that dissipates pH gradients across membranes, exhibited marked inhibition of GL uptake. *N,N’*-dicyclohexylcarbodiimide (DCCD; a H^+^-ATPase inhibitor), vanadate (P-type ATPase inhibitor), and nifedipine (a typical inhibitor of ABCB-type ABC transporters) showed a moderate inhibition of GL uptake, whereas ammonium chloride (NH_4_Cl), which eliminated the pH gradient across tonoplasts, did not inhibit GL uptake. Glutathione synthase inhibitor such as buthionine sulfoximine (BSO) and glutathione (GSH) did not affect GL uptake.

The uptake of glycyrrhetinic acid (GA), an aglycone of GL, by cells was similar to that of GL uptake. When GL and GA were added to the incubation medium, *G. glabra* cells preferentially incorporated GA, as shown in Figure 4. Glucuronic acid, a sugar moiety of GL, hardly inhibited GL uptake (data not shown).

### 2.3. GL Transport by Tonoplast Vesicles Obtained from Cultured Cells of G. glabra

Tonoplast vesicles were purified by sucrose density gradient centrifuge of microsomes prepared from cultured cells of *G. glabra*. Due to their activities, vanadate-sensitive ATPase and PPase were used as marker enzymes for plasma membrane and tonoplast vesicles, respectively. Tonoplast-rich vesicles were recovered from the interface between 0% and 20% sucrose layers (0–20% sucrose fraction), while plasma membrane-rich vesicles were recovered from the 30–40% sucrose fraction (Figure 5A,B). Membrane separation was confirmed by immunoblot analysis using antibodies against the luminal binding protein (BiP), all plasma membrane intrinsic proteins (all-PIP), and γ-tonoplast intrinsic protein (γ-TIP), which are marker proteins for the endoplasmic reticulum (ER), plasma membrane, and tonoplast, respectively (Figure 5C). GL uptake was the highest of the tonoplast-rich vesicles recovered from the 0–20% sucrose fraction (Figure 5D). Furthermore, GL transport was strictly dependent on MgATP and was not observed in its absence. The extent of GL uptake in the four fractions was consistent with γ-TIP levels in the fractions.

In addition, we examined the pH and temperature dependency of GL transport by tonoplast vesicles with Tris-MES buffer ranging from pH 6.5 to 9.0. As shown in Figure 6A, GL transport was unaffected by the pH of the incubation mixture. However, GL transport was temperature-dependent, with an optimal temperature of 45 °C (Figure 6B). GL uptake by tonoplast vesicles linearly increased within 10 min of the initiation of incubation and then gradually decreased, reaching a plateau within 30 min (Figure 7). Kinetic analysis indicated that GL transport by tonoplast vesicles exhibited Michaelis–Menten-type saturation kinetics with *K*_m_ values of 29 µM for GL and 1.2 mM for MgATP and a *V*_max_ of 6.1 pmol·min^−1^·mg protein^−1^ (Figure 8).

Various phosphate esters were used to examine their energization of GL transport by tonoplast vesicles (Table 1). MgATP was the most effective nucleotide triphosphate for GL uptake. MgGTP and MgUTP also supported GL transport, although their effects on MgATP were about 75% and 39%, respectively. MgAMP, MgADP, and pyrophosphate were ineffective for GL transport. 

### 2.4. Effects of Transport Inhibitors on GL Transport by Tonoplast Vesicles

We estimated the effects of various transport inhibitors on GL transport by tonoplast vesicles (Figure 9). Vanadate effectively suppressed GL uptake. CCCP, DCCP, and niflumic acid moderately inhibited the transport. However, bafilomycin A_1_, NH_4_Cl, and probenecid hardly inhibited GL transport.

### 2.5. Substrate Specificity of GL Transport by Tonoplast Vesicles

GL transport was estimated in the presence of GL-related compounds such as triterpene saponins and steroids to determine the substrate specificity of GL transport by tonoplast vesicles (Table 2). GA strongly competed with GL for uptake by the vesicles, and GA-3 monoglucuronide (3-MGA) also suppressed GL transport but to a lesser extent. Glucuronic acid did not affect the transport. Oleanane-type triterpenes (saikosaponin b_2_ and β-escin) but not dammarane-type saponin (ginsenoside R_e_) competed for GL transport. Steroid compounds did not significantly inhibit GL transport. The licoricesaponin G_2_ did not affect GL transport, despite its high similarity to the structure of GA.

## 3. Discussion

In this study, we demonstrated that the cultured cells of *G. glabra* efficiently incorporated exogenously supplied GL; however, GL uptake was not observed in cultured cells of *C. roseus* or *L. erythrorhizon*. GL uptake by the cultured cells may be related to the biosynthetic activity of GL in the original species of the cultured cells. In addition, such an association between the biosynthetic activity and incorporation potential for berberine was also reported suspension cultures for *Coptis japonic**a* [24]. However, dedifferentiated cells of *G. glabra* were not capable of producing GL, although the original plants accumulated GL in their roots and stolons [20], which is in sharp contrast to suspension cultures of *C. japonica* that actively synthesized berberine de novo. In fact, we could not detect GL or its aglycone GA in the cultured cells used in the present study. Thus, to the best of our knowledge, this is the first report indicating that the transport of the secondary metabolites is expressed even if the biosynthetic potential of the particular compounds is suppressed in the cultured cells.

The uptake of GL by cells apparently comprises two stages (Figure 10). GL is first incorporated into the cytoplasm through the plasma membrane and is then likely transported into the vacuoles through the tonoplast. This is suggested because the tonoplast-rich vesicles prepared from the cultured cells of *G. glabra* actively incorporated GL, although there was no direct evidence for the accumulation of GL in the vacuoles. Inhibitor experiments demonstrated that GL uptake by cells was drastically inhibited by probenecid and niflumic acid, whereas neither of the two inhibitors affected GL transport by tonoplast vesicles. Probenecid and niflumic acid are anion channel blockers and inhibit anion transport through membranes [25,26,27]. This indicates that an anion channel is involved in transport through plasma membranes but not in the tonoplast vesicles. In addition, CCCP strongly inhibited GL uptake by the cultured cells but only the intermediate inhibition of GL transport by the tonoplast membranes. Since CCCP is a protonophore that dissipates pH gradients across membranes [28,29], the ΔpH across the plasma membrane may play an important role in GL uptake by cells. Thus, a H^+^/GL symporter is present in the plasma membrane of *G. glabra* cells and could exogenously take up supplied GL into the cytoplasm by recognizing the anion site of the glucuronic acid moiety of GL. 

GL transport through tonoplast vesicles was dependent on MgATP and effectively inhibited by vanadate. Vanadate is a membrane ATPase inhibitor and represses the function of ABC transporters [30,31,32]. In contrast, neither niflumic acid nor probenecid affected GL transport by tonoplast vesicles. Bafilomycin A_1_ and NH_4_Cl also exhibited negligible inhibitory effects on GL transport through the tonoplast, indicating that the transport was insensitive to the transmembrane H^+^-electro potential difference [33,34]. Although GL transport by tonoplast vesicles was moderately inhibited by DCCD and CCCP, this may have been due to the presence of plasma membranes in the vesicle fraction. These results clearly indicate that GL transport by tonoplast vesicles is ABC-protein mediated. However, nifedipine, a specific inhibitor of the ABCB-type ABC transporter, did not inhibit GL transport by the tonoplast vesicles, suggesting that an ABC transporter other than the MDR (ABCB) type was involved in GL transport by the tonoplast vesicles of *G. glabra*.

Although there have been an increasing number of reports describing secondary metabolite transporting systems in plants, transporters of terpenoids remain unknown, except for sclareol, a diterpenoid defense compound produced by *Nicotiana* species, and crocin, an apocarotenoid pigment in *Crocus sativus*. From the cultured cells of *N. plumbaginifolia*, cDNA encoding a PDR-type ABC transporter, NpABC1, was cloned, and it was suggested that NpABC1 was located in the plasma membrane and involved in the secretion of sclareol in fungus infections [35]. On the other hand, the accumulation of crocin in the vacuoles of saffron stigmas was mediated by ABCC-type ABC transporters [36]. As with saffron, the present ABC transporter was expressed in the tonoplast of *G. glabra* cells and was involved in the inward transport of GL into the vacuoles. Sakai et al. and Shitan et al. characterized berberine (an alkaloid) transport by the cultured cells of *C. japonica* and found that an MDR-type ABC transporter (CjMDR1) was involved in the uptake of berberine through the plasma membrane into the cytoplasm [37,38]. Berberine is then transported into the vacuoles by H^+^-antiporters present in the tonoplast [39]. This is contrary to the GL transporting system in cells of *G. glabra,* where GL was first taken up by a H^+^-symporter located in the plasma membrane and then transported into vacuoles by an ABC transporter on the tonoplast.

The GL-transporting ABC-type protein in the tonoplast appears to strictly recognize the chemical structure of the aglycone moiety because GA strongly competed with GL for transport by the tonoplast vesicle, whereas licoricesaponin G_2,_ whose structure is similar to GL, did not affect transport. This result is also in contrast with the berberine transporting system in tonoplasts of *C. japonica*, where transport is competitively inhibited not only by other protoberberine alkaloids, such as palmatine and reticuline, but also by structurally unrelated alkaloids such as vinblastine. The broad specificity of berberine transport by the tonoplast may be due to the fact that it is mediated by a H^+^-antiporter protein and not ABC transporters [39]. In fact, a similar broad substrate specificity of a H^+^-antiporter was reported for a H^+^/flavove glucoside antiporter of barley tonoplasts [40].

In conclusion, exogenous GL was efficiently taken up by *G. glabra* cells. The process was species-specific, although the biosynthesis of GL was not expressed in the cultured cells. GL incorporation through the plasma membrane was mediated by a H^+^-symporter, whereas GL transport through the tonoplast was mediated by an ABC transporter, recognizing the strict chemical structure of the aglycone moiety. The molecular identification of GL transporters as well as their functional characterization in planta awaits further investigation.

## 4. Materials and Methods

### 4.1. Plant Materials and Chemicals

Cultured cells of *G. glabra* were maintained in Linsmaier-Skoog (LS) medium [41] containing 100 µM NAA and 1 µM BA subcultured at 2-week intervals in the dark at 25 °C. One-week-old cells were used for transport experiments unless otherwise stated. The cultured cells of *C. roseus* and *L. erythrorhizon* were continuously subcultured as previously described [42,43]. All chemicals used in this study were purchased from Wako Pure Chemicals (Osaka, Japan), Nacalai Tesque (Kyoto, Japan), and Merck (Darmstadt, Germany) unless otherwise stated.

### 4.2. GL Uptake by Cultured Cells

The cultured cells were aseptically harvested by filtration, and the cells (1.0 g fresh weight) were inoculated into 25 mL of fresh LS medium without NAA and BA. After adding GL at a final concentration of 100 µM, the cells were cultured on a rotary shaker at 25 °C. At 0, 1, 2, 4, and 6 h after GL addition, 90 µL aliquots of the medium were collected, mixed with 90 µL methanol, and centrifuged at 10,000× *g* for 10 min. The cells (0.1 g) were extracted with 1 mL of 80% (*v*/*v*) methanol by sonication. The methanol extracts were centrifuged at 10,000× *g* for 10 min. The supernatant was used for the quantitative determination of GL. 

### 4.3. Quantitation of GL

Samples were subjected to HPLC analysis under the following conditions: column, 5C_18_-AR-II waters 4.6 × 150 mm (Nacalai Tesque); mobile phase A, 0.1% formic acid; mobile phase B, acetonitrile; gradient program, 30–70% from 0 to 20 min—70–100% from 20 to 22 min, described as acetonitrile percent (time); flow rate, 1.0 mL/min; detection, 254 nm; column temperature, 40 °C. The amounts of GL were calculated based on a calibration curve prepared using the peak areas of standards (from 5 pmol to 200 pmol). 

### 4.4. Inhibitor Experiment with the Cultured Cells

Cultured cells of *G. glabra* were incubated with inhibitors in 25 mL of LS medium for 1 h before adding GL. According to another study [37], the final concentration of the compounds and the solvents used to prepare the stock solution (given in parentheses) are as follows: 100 µM NAA (water), 200 µM nifedipine (DMSO), 10 mM NH_4_Cl (water), 100 µM GA (DMSO), 500 µM glucuronic acid (water), 2 mM sodium orthovanadate (water), 200 µM probenecid (DMSO), 50 µM niflumic acid (DMSO), 10 µM DCCD (DMSO), 10 µM CCCP (DMSO), 1 mM BSO (water), and 100 µM GSH (water). Sodium orthovanadate was depolymerized before use according to the method of Goodno [44]. To 25 mL of LS medium, 50–125 µL of the stock solution was added. For the control, the same amount of DMSO or water was added. After adding GL (final concentration 100 µM), uptake was measured at 25 °C for 6 h.

### 4.5. Preparation of Tonoplast Vesicles 

Tonoplast vesicles were prepared from *G. glabra* cells by the method described by Otani et al. and Sugiyama et al. [39,45] with some modifications. All procedures were performed on ice or at 4 °C. Cells frozen with liquid N_2_ were homogenized in 3 mL/g of ice-cold homogenizing buffer containing 10% (*v*/*v*) glycerol, 0.5% (*w*/*v*) polyvinylpolypyrrolidone, 5 mM EDTA, 100 mM Tris-HCl (pH 8.0), 150 mM KCl, 3.3 mM dithiothreitol (DTT), and 1 mM phenylmethylsulfonyl fluoride (PMSF). The homogenate was strained through Miracloth (Merck) and centrifuged at 8000× *g* for 10 min. The supernatant was centrifuged again at 8000× *g* for 10 min, and the subsequent supernatant was centrifuged at 100,000× *g* for 40 min. The pellet was homogenized by 30 strokes of a Dounce homogenizer in resuspension buffer containing 10% (*v*/*v*) glycerol, 1 mM EDTA, 10 mM Tris-HCl (pH 7.6), 1 mM DTT, and 1 mM PMSF. The suspension was layered over a 20–30–40% (*w*/*v*) discontinuous sucrose gradient containing 1 mM EDTA, 10 mM Tris-HCl (pH 7.6), 1 mM DTT, and 1 mM PMSF in addition to sucrose and was centrifuged at 100,000× *g* for 120 min. Each fraction recovered from the interfaces was resuspended in resuspension buffer to dilute sucrose and centrifuged at 100,000× *g* for 40 min. Each pellet was homogenized in resuspension buffer to measure marker enzyme activities and GL transport. The membrane vesicles were stored at −80 °C until use.

### 4.6. Marker Enzyme Assay

The activity of vanadate-sensitive ATPase was measured as the marker enzyme for the plasma membrane vesicles, according to the method described by Yoshida et al. [46]. The total ATPase activity of the membrane vesicle was defined as 100%, and the activity reduced by the addition of vanadate was defined as the relative activity of the plasma membrane ATPase. The activity of PPase was measured as the marker enzyme for tonoplast vesicles by the method described by Maeshima and Yoshida [47].

### 4.7. Immunoblotting

The purity of vesicles was checked by immunoblotting with antibodies against all-PIP, γ-TIP, and ER luminal BiP. These antibodies were generously donated by Dr. Maeshima, Nagoya University. SDS-PAGE, transfer to polyvinylidene difluoride membranes, and subsequent immunodetection were performed as previously described [45].

### 4.8. Measurement of GL Transport with Tonoplast Vesicles

Spin columns for the transport assay were prepared by the method described by Otani et al. and Sugiyama et al. [39,45] with some modifications. First, small holes were punctured in the bottom and the lid of a 1.5 mL plastic tube with a needle (18 G). Next, siliconized glass wool was stuffed at the bottom of the prepared tube, and the tube was then placed in a new 1.5 mL plastic tube without holes. Sephadex G-50 Fine (GE Healthcare) suspended in 1.5 mL of 50 mM Tris-MES buffer (pH 7.5) was added to the top tube, followed by centrifugation at 400× *g* for 2 min twice, which resulted in the formation of a Sephadex G-50 gel bed of approximately 700 µL in the top tube.

Transport of GL by tonoplast vesicles was measured according to the method described by Otani et al. and Sugiyama et al. with some modifications [39,45]. The standard reaction mixture contained 50 mM Tris-MES (pH 7.5), 100 mM KCl, 5 mM MgATP, 100 µM GL, and membrane vesicles equivalent to 30 µg protein in a total volume of 200 µL unless otherwise specified. The stock solution of GL was prepared at a final concentration of 2 mM in water, and 10 µL was added per 200 µL of the assay mixture. Reactions were initiated by the addition of membrane vesicles. After incubation at 25 °C, 130 µL of the reaction mixture was loaded on the spin column and centrifuged at 400× *g* for 2 min at 4 °C. The filtrate was mixed with an equal volume of acetonitrile and sonicated for 10 min to extract GL from the membrane vesicles and centrifuged at 20,000× *g* for 10 min at 4 °C. The supernatants were subjected to HPLC analysis to quantitate GL.

### 4.9. Kinetic Analysis of GL Transport

The transport assay was performed with different GL concentrations (ranging from 12.5 to 200 µM) or different MgATP concentrations (from 0.25 to 4 mM). Reactions were performed at 25 °C for 5 min, and transported GL was measured as described above.

### 4.10. Inhibitor Experiment with Tonoplast Vesicles

The reaction mixture was prepared without MgATP, and each inhibitor was incubated with tonoplast vesicles for 2 min at 25 °C before adding MgATP. According to other studies [39,45], the final concentration of the transport inhibitors and the solvents used to prepare the stock solution (given in parentheses) are as follows: 1 mM sodium orthovanadate (water), 5 µM bafilomycin A_1_ (DMSO), 5 mM NH_4_Cl (water), 100 µM NAA (water), 100 µM GA (DMSO), 100 µM glucuronic acid (water), 100 µM nifedipine (DMSO), 100 µM probenecid (DMSO), 100 µM niflumic acid (DMSO), 100 µM DCCD (DMSO), and 100 µM CCCP (DMSO). To the 200 µL reaction mixture, 5 µL of each stock solution was added. In control, 5 µL of DMSO or water was added. For the competitive inhibition assay, GA (DMSO), 3-MGA (DMSO), licoricesaponin G_2_ (water:DMSO = 2:1, Tokiwa Phytochemical), saikosaponin b_2_ (DMSO), ginsenoside R_e_ (DMSO), β-escin (DMSO, MP Biomedicals), cholesterol (DMSO), stigmasterol (DMSO), ergosterol (DMSO), or cholic acid (water) was added to the reaction mixture at a concentration of 100 µM. After incubation at 25 °C for 5 min, transported GL was measured as described above. DMSO did not affect the GL transport at the concentrations used throughout this study.

## Figures and Tables

**Figure 1 plants-11-01250-f001:**
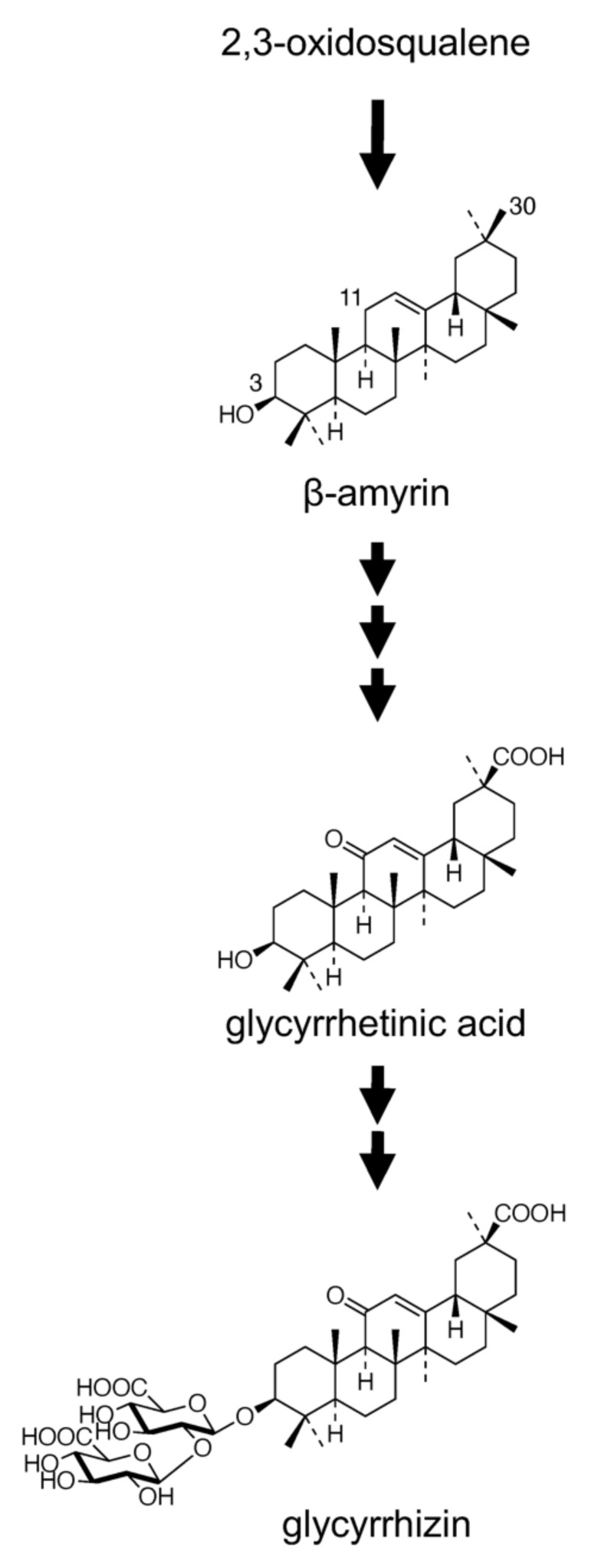
Proposed glycyrrhizin biosynthetic pathway.

**Figure 2 plants-11-01250-f002:**
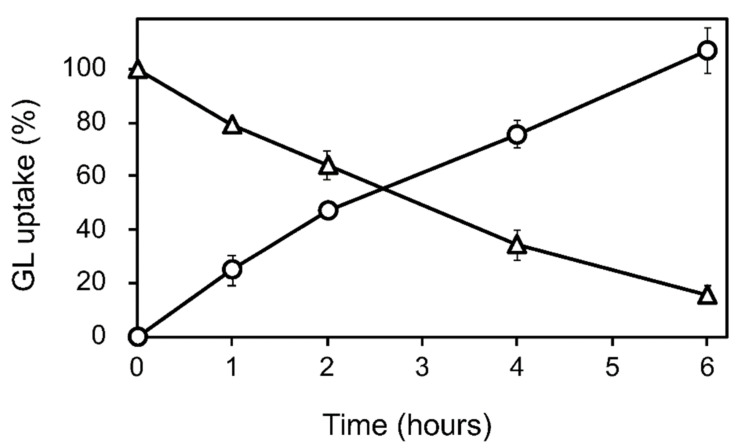
GL uptake by cultured cells of *G. glabra*. GL was added to the cultured cells at a concentration of 100 µM. Circles and triangles represent the contents of GL in the cell and the medium, respectively. The value of 100% indicates the complete uptake amounts of GL added to the medium. Data are presented as the mean ± standard deviation from triplicate cultures.

**Figure 3 plants-11-01250-f003:**
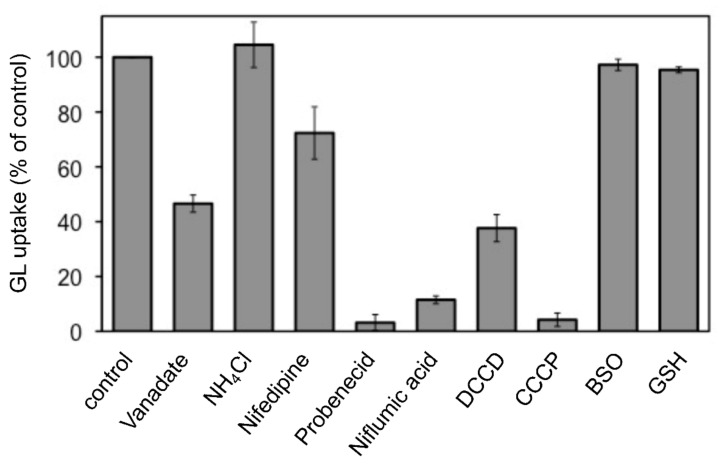
Effect of various transport inhibitors on GL uptake by cultured cells of *G. glabra*. Either nifedipine (200 µM), NH_4_Cl (10 mM), vanadate (2 mM), probenecid (200 µM), niflumic acid (50 µM), DCCD (10 µM), CCCP (10 µM), BSO (1 mM), or GSH (100 µM) was added 1 h before the addition of GL. GL uptake was measured 6 h after its addition (final concentration 100 µM). The value of 100% indicates the uptake amounts of GL without inhibitors. Data are presented as the mean ± standard deviation from triplicate cultures.

**Figure 4 plants-11-01250-f004:**
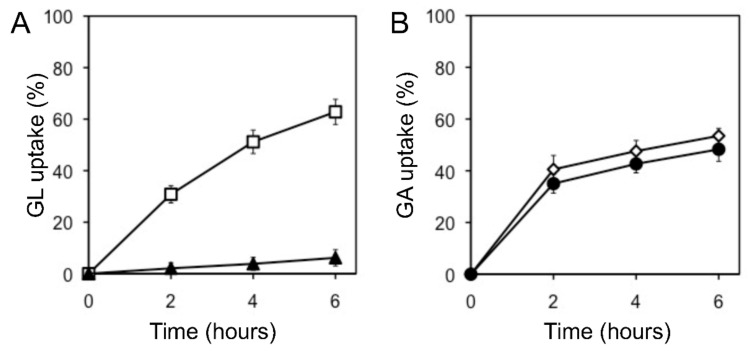
Effect of GA on GL uptake by cultured cells of *G. glabra*. (**A**) GL uptake was measured with (filled triangles) or without (open squares) GA. (**B**) GA uptake was measured with (filled circles) or without (open diamonds) GL. The final concentration of each compound was 100 µM. The value of 100% indicates the amounts of GL or GA added to the medium at 0 h. Data are presented as the mean ± standard deviation from triplicate cultures.

**Figure 5 plants-11-01250-f005:**
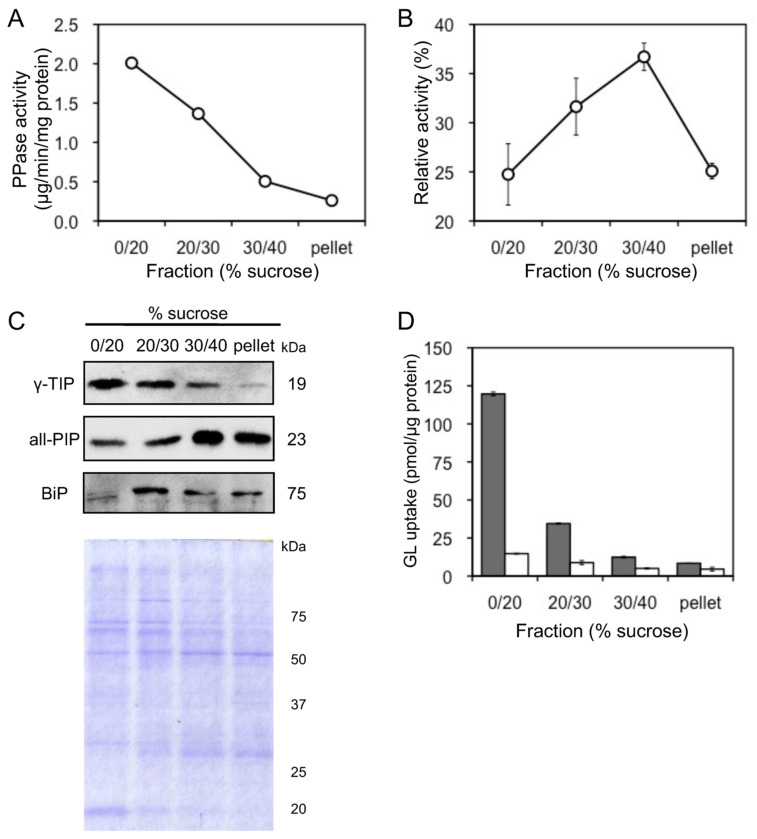
Uptake of GL by plasma membrane and tonoplast vesicles. (**A**) Activity of PPase was measured as a marker enzyme for tonoplast content. (**B**) Activity of vanadate-sensitive ATPase was measured as a marker enzyme for plasma membrane content. (**C**) BiP, all-PIP, and γ-TIP were immunodetected to confirm the purity of the ER, plasma membrane, and tonoplast vesicles. The SDS-PAGE gel was stained with Coomassie brilliant blue. (**D**) Each membrane fraction was incubated with 500 µM GL in the presence (grey) or absence (white) of 5 mM MgATP. Data are presented as the mean ± standard deviation from triplicate cultures.

**Figure 6 plants-11-01250-f006:**
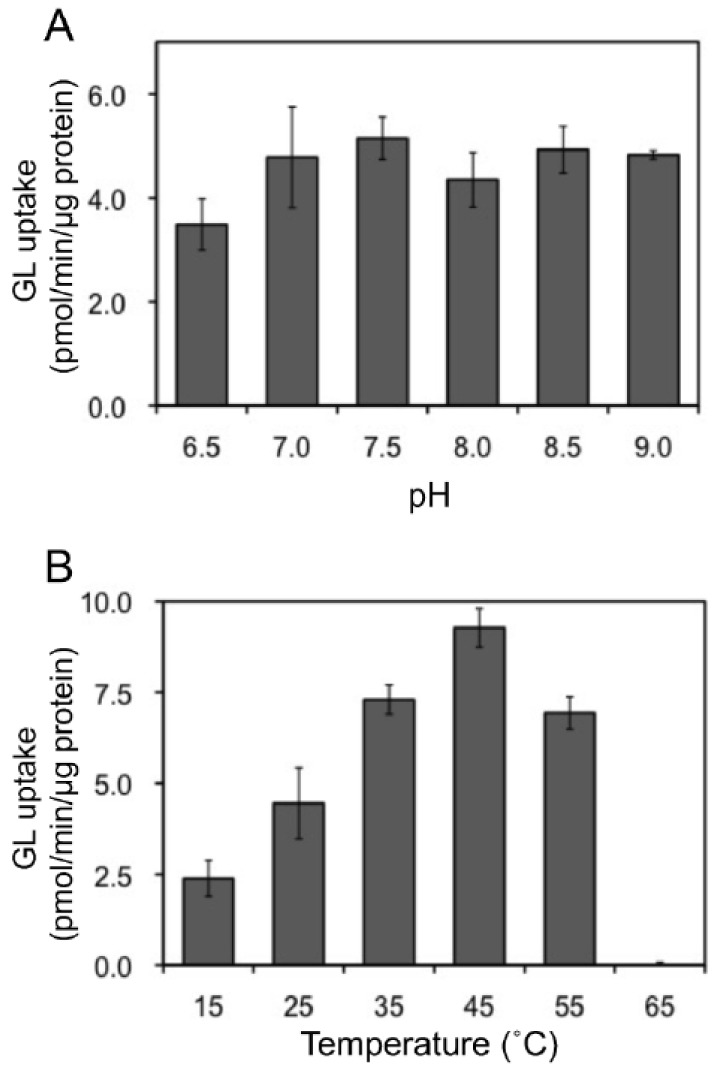
Effect of pH and temperature on ATP-dependent GL transport. The extent of GL transport was determined after 5 min of incubation with 100 µM GL. (**A**) Effect of pH on GL transport. The pH was adjusted with Tris-MES buffer. (**B**) The effect of temperatures on GL transport. Data are presented as the mean ± standard deviation from triplicate cultures.

**Figure 7 plants-11-01250-f007:**
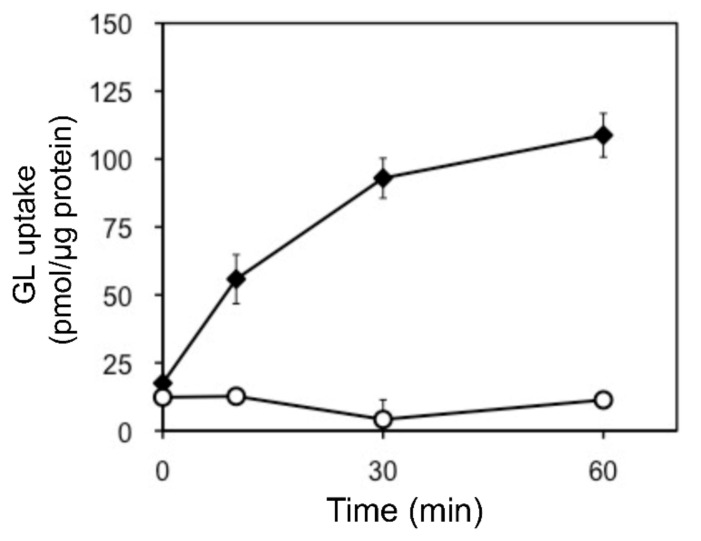
Time course of changes in GL uptake by tonoplast vesicles. Tonoplast vesicles (0–20% sucrose fraction) were incubated with GL (500 µM) in the presence (filled diamond) or absence (circle) of 5 mM MgATP. Data are presented as the mean ± standard deviation from triplicate cultures.

**Figure 8 plants-11-01250-f008:**
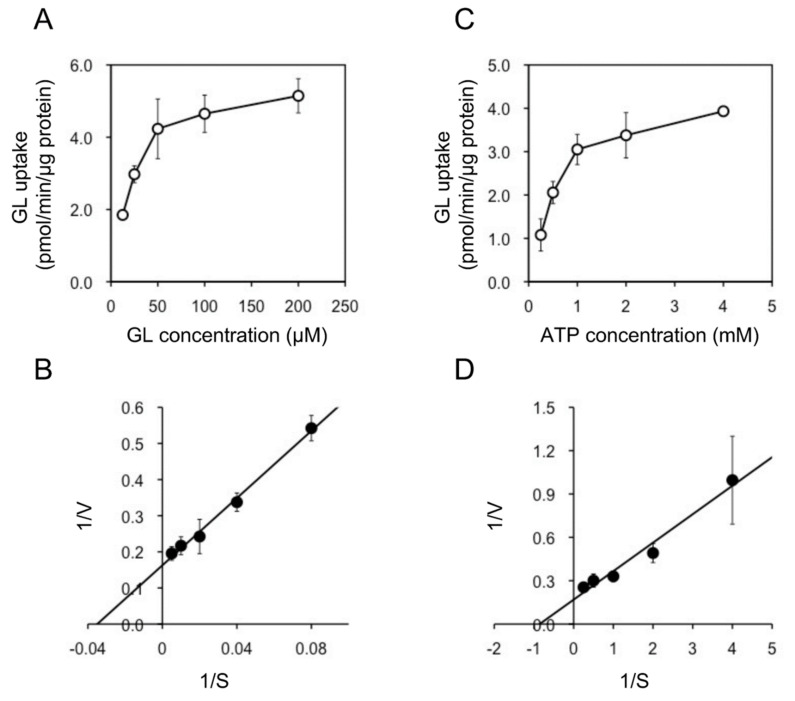
Kinetic analysis of ATP-dependent GL transport by tonoplast vesicles. (**A**) Tonoplast vesicles were incubated with various concentrations of GL in the presence of 5 mM MgATP. (**B**) Lineweaver–Burk plot of the concentration dependence of GL transport. (**C**) Tonoplast vesicles were incubated with 100 µM GL and various concentrations of MgATP. (**D**) Lineweaver–Burk plot of ATP-dependent GL transport. Data are presented as the mean ± standard deviation from triplicate cultures.

**Figure 9 plants-11-01250-f009:**
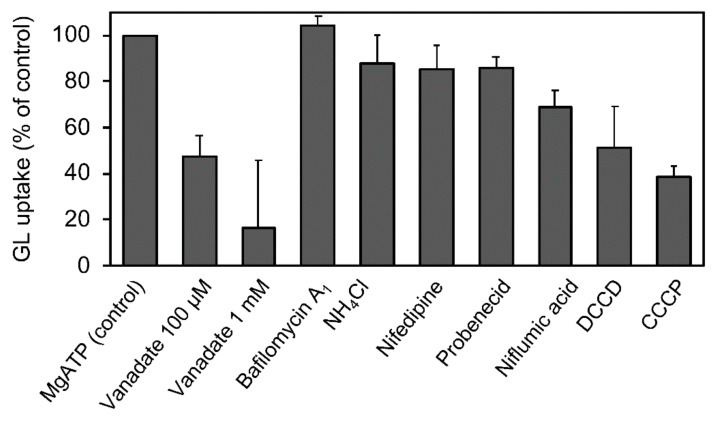
Effect of various transport inhibitors on ATP-dependent GL uptake. Tonoplast vesicles were incubated with 100 µM GL, and vanadate (1 mM), bafilomycin A_1_ (1 µM), NH_4_Cl (5 mM), nifedipine (100 µM), probenecid (100 µM), niflumic acid (100 µM), DCCD (100 µM), or CCCP (100 µM) was added before the addition of 5 mM of MgATP. The extent of GL transport was determined 5 min after incubation with 100 µM GL. The value of 100% indicates uptake of GL with MgATP and without inhibitors. Data are presented as the mean ± standard deviation from triplicate cultures.

**Figure 10 plants-11-01250-f010:**
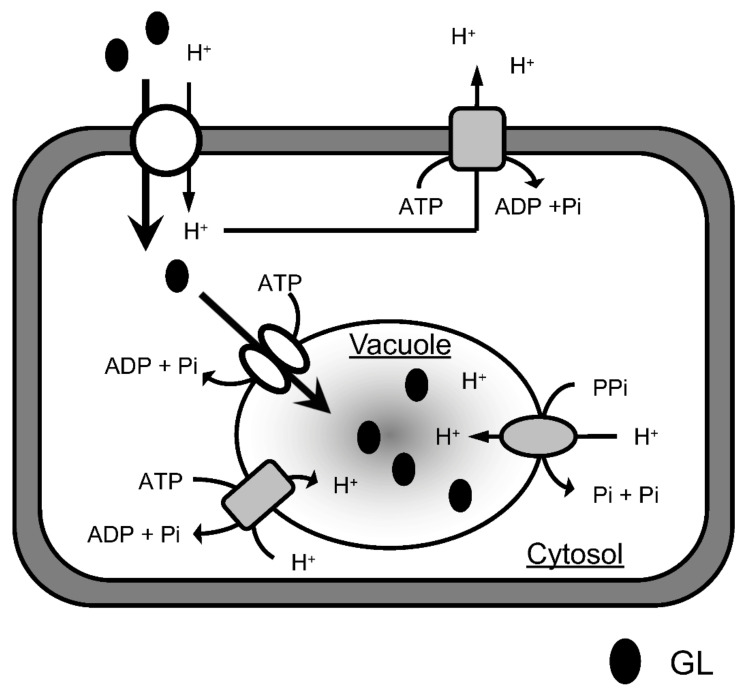
A proposed model for GL transport and accumulation. The H^+^/GL symporter slowly transports exogenous GL into the cytosol via a proton gradient formed by P-ATPase, while GL is rapidly taken up from the cytosol into the vacuole by the ABC transporter.

**Table 1 plants-11-01250-t001:** Effect of various nucleotides and pyrophosphate on GL uptake. Tonoplast vesicles were incubated with 100 µM GL in the presence of 5 mM of the compounds listed in the table. Values are presented as the mean ± standard deviation from triplicate cultures. N.D. = not detected.

Compound	GL Uptake (% of + MgATP)
–MgATP (negative control)	N.D
+MgPPi	N.D
+MgGTP	75.1 ± 2.7
+MgUTP	38.6 ± 16.0
+MgAMP	N.D
+MgADP	N.D

**Table 2 plants-11-01250-t002:** Effects of various compounds on ATP-dependent GL uptake. Tonoplast vesicles were incubated with 100 µM GL, and 5 mM MgATP in the absence (control) or presence of 100 µM of the compounds indicated. Values are presented as the mean ± standard deviation from triplicate cultures.

Compound	Glycyrrhizin Uptake (% of Control)
Glycyrrhetinic acid	15.6 ± 13.7
Glucuronic acid	102.3 ± 5.5
Glycyrrhetinic acid-3 monoglucuronide	65.4 ± 4.5
Licoricesaponin G_2_	96.3 ± 4.5
Saikosaponin b_2_	76.4 ± 17.0
Ginsenoside R_e_	97.7 ± 2.2
β-Escin	58.4 ± 8.3
Digitoxin	79.9 ± 7.3
Cholesterol	100.5 ± 4.0
Stigmasterol	112.2 ± 5.3
Ergosterol	93.7 ± 11.7
Cholic acid	103.2 ± 11.0

## Data Availability

The data presented in this study are available in the article.

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
