# Peer review of "Characterization of Triterpene Saponin Glycyrrhizin Transport by Glycyrrhiza glabra"

_plants, 2022, doi:10.3390/plants11091250_

Round 1

Reviewer 1 Report

The article concerns the study of a triterpene glycoside glycyrrhizin cell transport in Glycyrrhiza glabra cell culture. The glycoside consists of triterpene aglycone with a chromophore – conjugated keton group – and two residues of glucuronic acids that allows to control its contents in cells and media by HPLC with UV-detection. The biosynthesis of the glycoside is well studied but the mechanizm of its transportation was not studied and the cell culture of a plant-producer was used as a model for transport inhibitory analysis. A series of proton pump inhibitors including probenecid and niflumic acid, and a protonophore (carbonylcyanide m-chlorophenylhydrazone), significantly inhibited the triterpene glycoside input into cultured cells, while vanadate revealed moderate inhibition. The experiment concerning the glycoside transport by G. glabra tonoplast vesicles was depended not on proton-electrochemical gradient but MgATP and was significantly inhibited by vanadate. These observations clearly showed that the glycoside invasion into cultured cells is mediated by a proton-symporter in the plasma membrane and an ATP-binding cassette transporter into tonoblast vesicules. The last is high specific concerning the aglycone structure because even relative compounds could not be transported. GL on the tonoplast.

The article is very interesting and well written. However, there are several minor imperfections. I strictly recommend to provide normal structural formula of glycyrrhizin at the figure 1. The HPLC analytic procedure should be described more detail with presentation of the calibration curve. The phrase in the abstract “Despite the biosynthesis of GL in plants, little is known about GL transport and accumulation in plants” seems to be a little awkward and should be replaced with “In contrast to the biosynthesis of GL in plants, little is known about GL transport and accumulation in plants.”

Hence, the article may be published after minor corrections.

Author Response

Comments and Suggestions for Authors:

The article concerns the study of a triterpene glycoside glycyrrhizin cell transport in Glycyrrhiza glabra cell culture. The glycoside consists of triterpene aglycone with a chromophore – conjugated keton group – and two residues of glucuronic acids that allows to control its contents in cells and media by HPLC with UV-detection. The biosynthesis of the glycoside is well studied but the mechanism of its transportation was not studied and the cell culture of a plant-producer was used as a model for transport inhibitory analysis. A series of proton pump inhibitors including probenecid and niflumic acid, and a protonophore (carbonylcyanide m-chlorophenylhydrazone), significantly inhibited the triterpene glycoside input into cultured cells, while vanadate revealed moderate inhibition. The experiment concerning the glycoside transport by G. glabra tonoplast vesicles was depended not on proton-electrochemical gradient but MgATP and was significantly inhibited by vanadate. These observations clearly showed that the glycoside invasion into cultured cells is mediated by a proton-symporter in the plasma membrane and an ATP-binding cassette transporter into tonoplast vesicles. The last is high specific concerning the aglycone structure because even relative compounds could not be transported. GL on the tonoplast.

(Response)

Thank you very much for the positive evaluation. We have revised the manuscript according to the reviewers’ comments. Our responses are listed in a point-by-point manner as follows.

Point 1: I strictly recommend to provide normal structural formula of glycyrrhizin at the figure 1.

Response 1: The structural formula of glycyrrhizin was corrected in the revised Figure 1.

Point 2: The HPLC analytic procedure should be described more detail with presentation of the calibration curve.

Response 2: According to the comments, we have corrected and provided the detailed methods on lines 354–364.

Point 3: The phrase in the abstract “Despite the biosynthesis of GL in plants, little is known about GL transport and accumulation in plants” seems to be a little awkward and should be replaced with “In contrast to the biosynthesis of GL in plants, little is known about GL transport and accumulation in plants.”

Response 2: We have corrected the revised manuscript according to the comments.

Reviewer 2 Report

The manuscript entitled “Characterization of Triterpene Saponin Glycyrrhizin Transport by Glycyrrhiza glabra” written by the authors Kakuki Kato, Asako Horiba, Hiroaki Hayashi, Hajime Mizukami, Kazuyoshi Terasaka is devoted to the investigation of transport mechanism of triterpene glycoside glycyrrhizin in cultured cells of Glycyrrhiza glabra.

The research is well designed and characterized by sufficient level of novelty. However, it is not without some shortcomings.

The list of comments:

  1. Line 24: The phrase “Triterpene saponins are natural plant products” should be clarified, because these compounds are biosynthesized by some marine invertebrates (sea cucumbers and sponges) also.
  2. Line 41: “conjugating” should be replaced to “attachment/bonding”.
  3. Figure 5: The plots A and B should be inverted as they mentioned in the text. Fig. 5A – the vertical scale designation should be specified as it designed for Fig. 5B.
  4. Line 140: tonoplast intrinsic protein designated as (?-TIP), but in Fig. 5C – as g-TIP.
  5. Figure 8: C and D plots are not defined.
  6. Quantification of GL: The technique of evaluation of concentration of GL by HPLC is not described: whether calibration curve was constructed or the peak square was calculated? What unit of measurement is used for GL content (Figures 2, 3, 4 – vertical scale “GL uptake”)? It would be good to provide the HPLC profiles in Supporting materials.
  7. Lines 349 – 351, 413 – 415: How do the concentrations of inhibitors were selected? (These are different.)
  8. Line 380: Insert “The”

Author Response

(Comments and Suggestions for Authors)

The manuscript entitled “Characterization of Triterpene Saponin Glycyrrhizin Transport by Glycyrrhiza glabra” written by the authors Kakuki Kato, Asako Horiba, Hiroaki Hayashi, Hajime Mizukami, Kazuyoshi Terasaka is devoted to the investigation of transport mechanism of triterpene glycoside glycyrrhizin in cultured cells of Glycyrrhiza glabra.

(Response)

Thank you very much for the careful reading and the positive evaluation. We have revised the manuscript according to the reviewers’ comments. Our responses are listed in a point-by-point manner as follows.

Point 1: Line 24: The phrase “Triterpene saponins are natural plant products” should be clarified, because these compounds are biosynthesized by some marine invertebrates (sea cucumbers and sponges) also.

Response 1: Thank you very much for the detailed information. I agree with the reviewer. we corrected the following descriptions on line 24. “Triterpene saponins are widely distributed in plants and....”

Point 2: Line 41: “conjugating” should be replaced to “attachment/bonding”.

Response 2: We have corrected the revised manuscript according to the comments on line 41.

Point 3: Figure 5: The plots A and B should be inverted as they mentioned in the text. Fig. 5A – the vertical scale designation should be specified as it designed for Fig. 5B.

Response 3: The plots A and B in Figure 5 were inverted, and Figure legends were corrected (lines 157–159). The vertical scale of Figure 5A (plasma membrane ATPase activity) cannot be shown as Figure 5B (PPase activity) because the ATPase activity was measured by the relative activity based on absorbance without using a calibration curve. Thus, we added the following description to the Materials and Methods section on lines 398–400. “The total ATPase activity of the membrane vesicle was defined as 100%, and the activity reduced by the addition of vanadate was defined as the relative activity of the plasma membrane ATPase.”

Point 4: Line 140: tonoplast intrinsic protein designated as (?-TIP), but in Fig. 5C – as g-TIP.

Response 4: These are mistakes of type. We have corrected it as “γ-TIP” (lines 147, 148, 152, and 404).

Point 5: Figure 8: C and D plots are not defined.

Response 5: The labels C and D were added in the revised Figure 8.

Point 6: Quantification of GL: The technique of evaluation of concentration of GL by HPLC is not described: whether calibration curve was constructed or the peak square was calculated? What unit of measurement is used for GL content (Figures 2, 3, 4 – vertical scale “GL uptake”)? It would be good to provide the HPLC profiles in Supporting materials.

Response 6: According to the comments, we have corrected and provided the detailed methods on lines 354–364. GL uptake indicates the amount of uptake. Therefore, we have added "amounts" to the Figure legend (lines 95, 116, and 137–138). We also provided Figure S1 showing HPLC chromatograms corresponding to a portion of Figure 2.

Point 7: Lines 349 – 351, 413 – 415: How do the concentrations of inhibitors were selected? (These are different.)

Response 7: We have selected the concentrations of inhibitors according to other studies. Thus, we have added descriptions (lines 367 and 437–438).

Point 8: Line 380: Insert “The”

Response 8: We have corrected the revised manuscript according to the comments on line 403.

Reviewer 3 Report

This manuscript provides an excellent description and discussion of novel research. The experimental design covers a range of approaches to elucidate the mechanisms of uptake and intracellular transport of glycyrrhizin (GL) in cultured cells of Glycyrrhiza glabra.  The identification of the plasma membrane H+-symporter and tonoplast ABC transporter is novel.

While the biosynthesis of GL is well known, this work has provided great insight into the intracellular transport of GL and should enhance our understanding of the metabolism of other triterpenoids. The inclusion of the model in Figure 10 will assist in this information transfer to other researchers.

The manuscript is very well written. 

Author Response

(Comments and Suggestions for Authors)

This manuscript provides an excellent description and discussion of novel research. The experimental design covers a range of approaches to elucidate the mechanisms of uptake and intracellular transport of glycyrrhizin (GL) in cultured cells of Glycyrrhiza glabra.  The identification of the plasma membrane H+-symporter and tonoplast ABC transporter is novel.

While the biosynthesis of GL is well known, this work has provided great insight into the intracellular transport of GL and should enhance our understanding of the metabolism of other triterpenoids. The inclusion of the model in Figure 10 will assist in this information transfer to other researchers.

The manuscript is very well written.

(Response)

Thank you very much for the positive evaluation.